# Public attitudes in England towards the sharing of personal data following a mass casualty incident: a cross-sectional study

G James Rubin,[1] Rebecca Webster,[1] Antonia N Rubin,[2] Richard Amlot,[3] Nick Grey,[4] Neil Greenberg[1]

[1]Department of Psychological Medicine, King's College London, London, UK
[2]Tonbridge, Tonbridge, UK
[3]Behavioural Science Emergency Response Department Science and Technology (ERD S&T), Health Protection Directorate, Public Health England, Porton Down, UK
[4]Mood and Anxiety Clinical Academic Group, Sussex Partnership NHS Foundation Trust, Worthing, West Sussex, UK

**Correspondence to**
Dr G James Rubin;
Gideon.rubin@kcl.ac.uk

## ABSTRACT

**Objectives** To assess public attitudes towards data sharing to facilitate a mental health screening programme for people caught up in a mass casualty incident.

**Design** Two, identical, cross-sectional, online surveys, using quotas to ensure demographic representativeness of people aged 18–65 years in England. Participants were randomly allocated to consider a scenario in which they witness a terrorism-related radiation incident or mass shooting, after which a police officer records their contact details.

**Setting** Participants were drawn from an online panel maintained by a market research company. Surveys were conducted before and immediately after a series of terrorist attacks and a large tower block fire occurred in England.

**Participants** One thousand people aged 18–65 years participated in each survey.

**Main outcome measures** Three questions asking participants if it would be acceptable for police to share their contact details, without asking first, with 'a health-related government organisation, so they can send you a questionnaire to find out if you might benefit from extra care or support', 'a specialist NHS team, to provide you with information about ways to get support for any physical or mental health issues' and 'your GP, so they can check how you are doing'.

**Results** A minority of participants reported that it would be definitely not acceptable for their details to be shared with the government organisation (n=259, 13.0%), the National Health Service (NHS) (n=141, 7.1%) and their general practitioner (GP) (n=166, 8.3%). There was a small, but significant increase in acceptability for the radiation incident compared with the mass shooting. No major differences were observed between the preincident and postincident surveys.

**Conclusions** Although most people believe it is acceptable for their details to be shared in order to facilitate a mental health response to a major incident, care must be taken to communicate with those affected about how their information will be used.

## INTRODUCTION

Following a disaster, terrorist attack or other mass casualty incident, rates of mental distress

### Strengths and limitations of this study

► The survey sample was demographically representative of people aged 18–65 years in England.
► The survey was conducted again after the recent terrorist incidents and the Grenfell tower fire as we felt that the scenarios may have become more salient for respondents, providing a better test of attitudes towards data sharing in such contexts.
► Although demographically representative, our sample volunteered to answer internet surveys and may be more accepting of data sharing than those who did not volunteer.
► The scenarios used in our surveys were hypothetical. Being directly involved in a major incident may alter someone's views about the desirability of data sharing.
► Participants were asked to consider that neither they nor anyone they knew well had been harmed in the described scenarios. This may have reduced the perceived need to participate in a psychological screening programme and hence the perceived acceptability of data sharing.

and disorder among survivors, their relatives, eyewitnesses and first responders can be substantial.[1] Although evidence-based interventions are available to treat established mental health disorder, failure to seek professional help for mental illness is common as a result of stigma and other barriers to care.[2 3] Because of this, in their guidance on the detection and treatment of post-traumatic stress disorder, the UK National Institute for Health and Care Excellence recommends that 'screening of all [affected] individuals should be considered by the authorities responsible for developing the local disaster plan'.[4] Within the UK, the first use of a 'screen and treat' approach occurred following the 7 July 2005 bombings in London. A central team used a range of methods to contact as many people as possible who had been

caught up in the incident. Those contacted were asked to complete a short mental health questionnaire, invited to have a more detailed assessment if need be, and referred for treatment where required.[5] Although the process resulted in many patients receiving treatment who might not otherwise have done so, difficulties were encountered in assembling a comprehensive list of people to be screened. Sharing of information between agencies was hampered by a cautious interpretation of legislation designed to safeguard personal data.[6] As a result, only 910 people were contacted out of an estimated 4000 who were exposed to the attacks. Within the UK, greater clarity about the flexibility that responders have to share information was subsequently provided.[6] This guidance notes that where information is not 'sensitive' (ie, related to ethnicity, political opinion, religious belief, trade union membership, health, sexual life or criminal activity) then it is acceptable for names and contact details to be shared between official agencies to facilitate the provision of care to those affected.

In June 2015, 38 people were killed by a terrorist in Sousse, Tunisia. Thirty of those killed were British holidaymakers. Given the large number of British tourists who witnessed the attacks and felt their own lives to be in danger, a psychological health screening programme was again set up and subsequently widened to include victims of terrorist attacks in Brussels and Paris. Public Health England was given the task of setting up a registry of all English nationals who were directly affected by the attacks, something which might also be considered in the future for other forms of disaster.[7] Once again, however, obtaining the contact details for many of those affected was not possible. Although the police collected contact details from holidaymakers as they returned to England, legal advice prevented them from sharing this information. Difficulty in sharing information between agencies often hampers response efforts following a major incident. While it is sometimes possible to find more indirect routes to contact people, these approaches are often less than ideal. For example, while it might be possible for the police to send out information to people on behalf of a screening service or to ask for permission before sharing data, in practice, such strategies can be time consuming to agree, often result in a poorer response rate and prevent any single agency from combining overlapping lists of names held by different groups into a single, consolidated list of those affected.

Clearly, uncertainties still exist about when it is appropriate to share information following a major incident. Changes to legislation in 2018 will create stiffer penalties for organisations found to be in breach of data protection law and are likely to create an even more cautious approach to data sharing. But how members of the public themselves view this matter is unclear. Recent polling suggests that public attitudes towards data sharing are shaped by who is doing the sharing, what for and whether safeguards are in place.[8] Data sharing for the public good is generally viewed positively. For example, only 9% of

respondents to one large UK survey felt that it would be an invasion of privacy if details held about them on a cancer registry were used to invite them to participate in a study conducted by a university medical school.[9] Where a personal benefit is possible, people also tend to be more willing for information to be shared.[10] To our knowledge there has been no research on the public acceptability of data sharing in the context of a major public health incident, disaster or terrorist attack.

In this study, we used an online quota survey of a sample of adults in England designed to reflect the known population profile, to identify public attitudes towards the sharing of personal information following a mass casualty incident and the setting up of a register of affected people. We tested whether attitudes might differ depending on whether the incident resulted primarily in a mental or physical health risk for those involved in the incident. Soon after we conducted our survey, four terrorist attacks and a tower block fire occurred in England killing over 100 people. Given the intense media attention about the need to provide support to victims of these incidents, we felt that the scenarios described in our survey may have become more salient for respondents, providing a better test of attitudes towards data sharing in such contexts. We therefore repeated the survey with a new sample of respondents.

## METHOD

### Design

We commissioned the market research organisation Ipsos MORI to conduct two online surveys of 1000 adult (18–65) residents of England. Data collection for the initial, preincident survey took place between 9 and 15 March 2017. Four terrorist attacks then occurred in England between 22 March and 19 June, killing 34, and a tower block fire occurred on 14 June, killing 71. Data collection for our second, postincident survey took place between 3 and 10 July 2017. The procedures and questionnaires for both surveys were identical.

### Participants

Ipsos MORI recruited participants from an existing panel of people willing to take part in internet surveys (approximate n=160 000). Panel participants typically receive points for every survey they complete. For our surveys, participants received points equivalent to 30 pence. Quotas based on participant age and gender (interlocked), location and working status were used to ensure that the sample was demographically representative of adults aged 18–65 years in England, according to data from the National Readership Survey.[11] We excluded adults aged over 65 years on the basis that older adults who are members of an internet survey panel may not be representative of the general population of older adults.[12] We intended to recruit 1000 participants for each survey to provide us with a maximum sample error of about ±3%.

## Scenarios

Ipsos MORI emailed a link to the survey to potential participants. After providing informed consent and clicking through to begin the survey, participants received questions about one of two hypothetical scenarios. Which scenario they received was decided by the survey software, based on which scenario had the lowest number of completed responses at that time. We based the scenarios on an example given in HM Government's 'Data Protection and Sharing—Guidance for Emergency Planners and Responders'[6] (case study 5). The two versions represented an incident that might primarily pose a mental health threat (witnessing shooting) or a physical health threat (exposure to radiation). In the first version, participants were asked to 'imagine that you are on holiday in another country and witness a terrorist shooting. Some people are badly injured but you are not harmed and neither is anyone you know well'. The second version asked them to 'imagine that you are on holiday in another country and a place you visit is discovered by police to be contaminated by radioactive material. The police believe this is linked to a terrorist group. Some people are badly injured but you are not harmed and neither is anyone you know well'. Participants in both versions were then told that 'the British government arranges for you to be flown home, along with other British nationals who were in the area. When you arrive back in Britain, a police officer at the airport records your name, address, phone number and email address. The following questions ask about things that might happen next'.

## Questionnaire

The supplementary materials show the full text for all items. We first presented participants with eight groups who might ask the police to share information with them following the incident, together with their reasons for this. In each case, we asked whether the participant thought the police currently could share information with this group without asking the participant first, and whether the participant thought it was acceptable for the police to share their information with the group.

We next provided information about the rationale and process for a mental health screen and treat programme. We described it as a specialist NHS service set-up to offer support and treatment to those who need it and explained that a government organisation would support the service by putting together a confidential database of those affected and writing to them with a screening questionnaire. We asked participants to state their level of agreement with the statements that ' If I was caught up in a terrorist incident, I would want the police to give my contact details to the government organisation so that they can contact me about this service;' 'I would want my name to be included on this database;' and 'I would be unhappy if my name was included on this database.'

Participants were then asked to reconsider the scenario they had been presented with and told that the police had decided to share their name and contact details with a health-related government organisation. They were asked to state their agreement with 10 statements concerning their views about this (eg, 'it would be an invasion of my privacy').

We asked all participants about their age, gender, working status, where in England they lived and their highest educational qualification.

## Analysis

We calculated the proportion of respondents endorsing each response option and assessed the difference in responses between the preincident and postincident surveys and between the two scenarios using $\chi^2$ tests.

## Patient involvement

No patients or members of the public were involved in the design, recruitment or conduct of the study. Due to anonymity, results cannot be disseminated to study participants unless they specifically request them from the researcher.

## RESULTS

The top-line results provided by Ipsos MORI for both surveys can be found in online supplementary files 1 and 2.

## Response rates and demographics

Response rates to the invitation emails were 10% (preincident) and 6% (postincident). These are normal rates for surveys of this nature. The lower response rate for the postincident survey may reflect the fact that data collection occurred during July which coincides with the summer holiday period in England. In each case, a small proportion of respondents (45 in total) were excluded from the data for completing the surveys suspiciously quickly or giving illogically identical answers to multiple consecutive questions. Demographics are shown in table 1.

## Perceived ability and acceptability of the police sharing data with other agencies

Table 2 presents data on the perceived ability of the police to share personal data with other agencies. The option that was most commonly believed to be within the police's current powers was sharing data with 'a specialist NHS team, to provide you with information about ways to get support for any physical or mental health issues' (endorsed by 60.8% overall). 53.9% of respondents believed that police could definitely or probably share data with 'your GP, so they can check how you are doing', while 45.9% believed that data could definitely or probably be shared with 'a health-related government organisation, so they can send you a questionnaire to find out if you might benefit from extra care or support'. No differences were found between the preincident and postincident surveys. Significantly more people believed that data could be shared following a radiation incident compared with a shooting incident with: your general practitioner

**Table 1** Demographic characteristics of survey respondents

| Variable | Categories | Frequency (%) | | |
| --- | --- | --- | --- | --- |
| | | **Preincidents** | **Postincidents** | **Total** |
| Sex | Male | 490 (49) | 496 (49.6) | 986 (49.3) |
| | Female | 510 (51) | 504 (50.4) | 1014 (50.7) |
| Age | 18–24 | 119 (11.9) | 144 (14.4) | 263 (13.2) |
| | 25–34 | 225 (22.5) | 222 (22.2) | 447 (22.4) |
| | 35–44 | 213 (21.3) | 207 (20.7) | 420 (21) |
| | 45–54 | 233 (23.3) | 226 (22.6) | 459 (23) |
| | 55–65 | 210 (21) | 201 (20.1) | 411 (20.6) |
| Region | North East | 46 (4.6) | 46 (4.6) | 92 (4.6) |
| | North West | 137 (13.7) | 135 (13.5) | 272 (13.6) |
| | Yorkshire and Humberside | 98 (9.8) | 101 (10.1) | 199 (10) |
| | West Midlands | 97 (9.7) | 101 (10.1) | 198 (9.9) |
| | East Midlands | 89 (8.9) | 84 (8.4) | 173 (8.7) |
| | East of England | 90 (9) | 99 (9.9) | 189 (9.5) |
| | South West | 95 (9.5) | 90 (9) | 185 (9.3) |
| | South East | 167 (16.7) | 166 (16.6) | 333 (16.7) |
| | London | 181 (18.1) | 178 (17.8) | 359 (18) |
| Employment status | Working full time | 501 (50.1) | 509 (50.9) | 1010 (50.5) |
| | Working part time | 151 (15.1) | 140 (14) | 291 (14.6) |
| | Self-employed | 70 (7) | 78 (7.8) | 148 (7.4) |
| | Unemployed, looking for a job | 49 (4.9) | 52 (5.2) | 101 (5.1) |
| | Unemployed not looking for a job | 105 (10.5) | 103 (10.3) | 208 (10.4) |
| | Retired | 78 (7.8) | 56 (5.6) | 134 (6.7) |
| | Pupil/student/in full-time education | 46 (4.6) | 62 (6.2) | 108 (5.4) |
| Highest level of education achieved | Left school without qualifications | 33 (3.3) | 27 (2.7) | 60 (3) |
| | Secondary education | 424 (42.4) | 413 (41.3) | 837 (41.9) |
| | Higher education | 538 (53.8) | 555 (55.5) | 1093 (54.7) |
| | Prefer not to say | 5 (0.5) | 5 (0.5) | 10 (0.5) |

(GP), a health-related government organisation and a specialist NHS team.

Table 3 presents data on the acceptability of the police sharing personal data. The option that was most commonly seen as being acceptable was sharing data with a specialist NHS team (seen as definitely acceptable by 27.6% and probably acceptable by 44.3%). Most respondents believed it was definitely (29.9%) or probably (40.2%) acceptable for the police to share data with their GP, while a smaller number believed it was definitely (13.0%) or probably (37.6%) acceptable for data to be shared with a health-related government organisation. No differences were found between the preincident and postincident surveys, aside from a small reduction in the postincident data in the acceptability of information being shared with medical researchers. Significantly more people believed it would be acceptable to share their data following a radiation incident compared with a shooting incident with: their GP, a team of university

medical researchers and a health-related government organisation.

### Attitudes towards data sharing to enable a screen and treat programme or database

Table 4 presents the attitudes of participants towards a screen and treat programme and confidential database. Although most respondents strongly agreed (21.0%) or tended to agree (45.6%) that they would want the police to share their contact details to allow them to be contacted about screening, 10.7% tended to disagree and 5.2% strongly disagreed. In total, 51.6% strongly agreed or tended to agree with wanting their name to appear on the database, while 18.3% tended to agree that they would be unhappy if it was included on the database and 9.6% strongly agreed that they would be unhappy. No differences were found between the preincident and postincident surveys. Because these questions asked about 'a terrorist incident' in general, we did not

**Table 2** Ability of police sharing personal data with other agencies, without asking first

| Agency requesting personal information and reason for the request | | Frequency (%) believing if police are able to share for this reason | | | | | | |
|---|---|---|---|---|---|---|---|---|
| | | Shooting | Radiation | $\chi^2$ | Preincidents | Postincidents | $\chi^2$ | Total |
| Your GP, so they can check how you are doing | Definitely able | 169 (16.9) | 208 (20.8) | 20.521*** | 181 (18.1) | 196 (19.6) | 1.191 | 377 (18.9) |
| | Probably able | 328 (32.8) | 373 (37.3) | | 360 (36) | 341 (34.1) | | 701 (35.1) |
| | Not sure | 212 (21.2) | 203 (20.3) | | 206 (20.6) | 209 (20.9) | | 415 (20.8) |
| | Probably not able | 184 (18.4) | 122 (12.2) | | 154 (15.4) | 152 (15.2) | | 306 (15.3) |
| | Definitely not able | 107 (10.7) | 94 (9.4) | | 99 (9.9) | 102 (10.2) | | 201 (10.1) |
| Your travel insurance company, so they can offer you practical and financial support | Definitely able | 97 (9.7) | 93 (9.3) | 1.553 | 84 (8.4) | 106 (10.6) | 4.440 | 190 (9.5) |
| | Probably able | 235 (23.5) | 230 (23) | | 236 (23.6) | 229 (22.9) | | 465 (23.3) |
| | Not sure | 254 (25.4) | 263 (26.3) | | 271 (27.1) | 246 (24.6) | | 517 (25.9) |
| | Probably not able | 242 (24.2) | 226 (22.6) | | 236 (23.6) | 232 (23.2) | | 468 (23.4) |
| | Definitely not able | 172 (17.2) | 188 (18.8) | | 173 (17.3) | 187 (18.7) | | 360 (18) |
| A team of medical researchers from a university, so they can invite you to take part in a study to improve the way future incidents are dealt with | Definitely able | 38 (3.8) | 57 (5.7) | 7.437 | 51 (5.1) | 44 (4.4) | 1.220 | 95 (4.8) |
| | Probably able | 123 (12.3) | 144 (14.4) | | 135 (13.5) | 132 (13.2) | | 267 (13.3) |
| | Not sure | 226 (22.6) | 232 (23.3) | | 221 (22.1) | 237 (23.7) | | 458 (22.9) |
| | Probably not able | 314 (31.4) | 296 (29.6) | | 309 (30.9) | 301 (30.1) | | 610 (30.5) |
| | Definitely not able | 299 (29.9) | 271 (27.1) | | 284 (28.4) | 286 (28.6) | | 570 (28.5) |
| A journalist, so they can write a news article about the incident | Definitely able | 27 (2.7) | 21 (2.1) | 2.929 | 25 (2.5) | 23 (2.3) | 1.418 | 48 (2.4) |
| | Probably able | 52 (5.2) | 65 (6.5) | | 54 (5.4) | 63 (6.3) | | 117 (5.9) |
| | Not sure | 132 (13.2) | 142 (14.2) | | 133 (13.3) | 141 (14.1) | | 274 (13.7) |
| | Probably not able | 203 (20.3) | 206 (20.6) | | 202 (20.2) | 207 (20.7) | | 409 (20.4) |
| | Definitely not able | 586 (58.6) | 566 (56.6) | | 586 (58.6) | 566 (56.6) | | 1152 (57.6) |
| A charity such as the British Red Cross, so they can offer you support | Definitely able | 55 (5.5) | 62 (6.2) | 1.593 | 60 (6) | 57 (5.7) | 3.079 | 117 (5.9) |
| | Probably able | 201 (20.1) | 193 (19.3) | | 183 (18.3) | 211 (21.1) | | 394 (19.7) |
| | Not sure | 281 (28.1) | 272 (27.2) | | 281 (28.1) | 272 (27.2) | | 553 (27.7) |
| | Probably not able | 266 (26.6) | 285 (28.5) | | 286 (28.6) | 265 (26.5) | | 551 (27.6) |
| | Definitely not able | 197 (19.7) | 188 (18.8) | | 190 (19) | 195 (19.5) | | 385 (19.3) |
| A health-related government organisation, so they can send you a questionnaire to find out if you might benefit from extra care or support | Definitely able | 95 (9.5) | 116 (11.6) | 21.036*** | 104 (10.4) | 107 (10.7) | 0.532 | 211 (10.6) |
| | Probably able | 314 (31.4) | 393 (39.3) | | 358 (35.8) | 349 (34.9) | | 707 (35.4) |
| | Not sure | 283 (28.3) | 249 (24.9) | | 266 (26.6) | 266 (26.6) | | 532 (26.6) |
| | Probably not able | 185 (18.5) | 147 (14.7) | | 161 (16.1) | 171 (17.1) | | 332 (16.6) |
| | Definitely not able | 123 (12.3) | 95 (9.5) | | 111 (11.1) | 107 (10.7) | | 218 (10.9) |
| A specialist NHS team, to provide you with information about ways to get support for any physical or mental health issues | Definitely able | 168 (16.8) | 205 (20.5) | 22.844*** | 179 (17.9) | 194 (19.4) | 2.120 | 373 (18.7) |
| | Probably able | 395 (39.5) | 448 (44.8) | | 429 (42.9) | 414 (41.4) | | 843 (42.1) |
| | Not sure | 218 (21.8) | 195 (19.5) | | 209 (20.9) | 204 (20.4) | | 413 (20.7) |
| | Probably not able | 146 (14.6) | 89 (8.9) | | 111 (11.1) | 124 (12.4) | | 235 (11.8) |
| | Definitely not able | 73 (7.3) | 63 (6.3) | | 72 (7.2) | 64 (6.4) | | 136 (6.8) |

**Table 2** Continued

| Agency requesting personal information and reason for the request | | Frequency (%) believing if police are able to share for this reason | | | | | | |
|---|---|---|---|---|---|---|---|---|
| | | Shooting | Radiation | $\chi^2$ | Preincidents | Postincidents | $\chi^2$ | Total |
| A law firm, so they can offer to represent you in a 'no-win, no-fee' claim for compensation | Definitely able | 41 (4.1) | 33 (3.3) | | 36 (3.6) | 38 (3.8) | | 74 (3.7) |
| | Probably able | 87 (8.7) | 94 (9.4) | 6.087 | 91 (9.1) | 90 (9) | 0.121 | 181 (9.1) |
| | Not sure | 164 (16.4) | 197 (19.7) | | 182 (18.2) | 179 (17.9) | | 361 (18.1) |
| | Probably not able | 245 (24.5) | 253 (25.3) | | 247 (24.7) | 251 (25.1) | | 498 (24.9) |
| | Definitely not able | 463 (46.3) | 423 (42.3) | | 444 (44.4) | 442 (44.2) | | 886 (44.3) |

***P<0.001.
GP, general practitioner; NHS, National Health Service.

test for differences between the shooting and radiation scenarios.

Table 5 presents the broader attitudes of participants towards data-sharing in the context of the radiation and shooting scenarios. Concerns endorsed by more than 50% of respondents about the police sharing their information were that: my details would be made public by accident, I would be concerned about how my information might be used in the future and I would be concerned that my details might be shared by the health organisation with other groups without my permission. Most participants reported wanting to be kept informed about how their information was being used. More positively, 55.4% strongly agreed or tended to agree that they would be 'reassured that a health organisation was looking out for me' and 54.4% would trust the health organisation to keep their details secure. No differences were found between and preincident and postincident surveys. Significantly more people in the radiation than the shooting scenario reported that they would be reassured that a health organisation was looking out for them and also that they would want to be kept informed about how their information was being used. Significantly fewer people in the radiation scenario reported that the sharing would be an invasion of their privacy.

## DISCUSSION

Our data suggest that not only do the majority of people believe that it is acceptable for the police to share their details with health agencies in the aftermath of a major incident, but that most people believe that the police are currently able to do this under existing legislation. This belief is not simply due to a broader misunderstanding among the public about their data protection rights: only 8% believe that the police are able to share data with journalists, for example. Instead, in line with findings reported by others,[10] it appears that where a direct benefit to the individual might result from sharing information with a trusted source such as a GP or NHS service, the public have a greater acceptability of it, and an expectation that it will occur. However, such views were not unanimous among our participants. It was notable that 7% of our

sample thought it was definitely not acceptable for their details to be shared with an NHS service following a major incident, 8% thought it was definitely not acceptable for their details to be shared with their GP and 10% strongly agreed that they would be unhappy for their name to be included on a confidential database of those affected. More people (8%, 9% and 18%, respectively) thought these actions were probably not acceptable or tended to agree that they would be unhappy. These figures must be taken into consideration by responding agencies. Given that it is impossible to know in advance who is or is not willing for his/her data to be shared, not sharing data without prior permission to protect the interests of this minority may be a justifiable position in some cases.

Our data provide some indications as to why people may be concerned about data sharing following an incident. In particular, the risk of accidental or deliberate sharing of information to a third party was noted as a concern by many. If data sharing or the setting up of a database is to be considered following any future incident, a robust policy around further sharing should be developed and explained to those affected. Keeping people regularly updated on how their data are being used is also important. More generally, the nature of the incident itself is a determinant of the acceptability of data sharing and the use of a database. Participants who were asked to imagine a scenario that might pose a long-term physical health risk to themselves (eg, possible exposure to radiation) were more likely to have a positive view of data sharing than those who were asked to consider a risk to their mental health (eg, witnessing a shooting). Whether this was because participants found it hard to envisage that their mental health might be affected in the scenarios that we used, or whether this reflects the broader stigma associated with mental illness is unclear. Similarly, while people may be familiar with the concept of physical health effects that develop some time later after a radiation incident, they may be less familiar with the concept of mental health effects that only become apparent later: this lack of understanding may also have reduced the perceived desirability of a mental health screening programme across both conditions. It is

**Table 3** Acceptability of police sharing personal data with other agencies, without asking first

| Agency requesting personal information and reason for the request | | Frequency (%) believing if it is acceptable for the police to share for this reason | | | | | | |
|---|---|---|---|---|---|---|---|---|
| | | Shooting | Radiation | $\chi^2$ | Preincidents | Postincidents | $\chi^2$ | Total |
| Your GP, so they can check how you are doing | Definitely acceptable | 274 (27.4) | 324 (32.4) | | 297 (29.7) | 301 (30.1) | | 598 (29.9) |
| | Probably acceptable | 400 (40) | 404 (40.4) | 10.352* | 394 (39.4) | 410 (41) | 3.490 | 804 (40.2) |
| | Not sure | 131 (13.1) | 120 (12) | | 121 (12.1) | 130 (13) | | 251 (12.6) |
| | Probably not acceptable | 99 (9.9) | 82 (8.2) | | 101 (10.1) | 80 (8) | | 181 (9.1) |
| | Definitely not acceptable | 96 (9.6) | 70 (7) | | 87 (8.7) | 79 (7.9) | | 166 (8.3) |
| Your travel insurance company, so they can offer you practical and financial support | Definitely acceptable | 107 (10.7) | 114 (11.4) | | 113 (11.3) | 108 (10.8) | | 221 (11.1) |
| | Probably acceptable | 291 (29.1) | 271 (27.1) | 2.649 | 274 (27.4) | 288 (28.8) | 3.338 | 562 (28.1) |
| | Not sure | 214 (21.4) | 222 (22.2) | | 232 (23.2) | 204 (20.4) | | 436 (21.8) |
| | Probably not acceptable | 175 (17.5) | 160 (16) | | 158 (15.8) | 177 (17.7) | | 335 (16.8) |
| | Definitely not acceptable | 213 (21.3) | 233 (23.3) | | 223 (22.3) | 223 (22.3) | | 446 (22.3) |
| A team of medical researchers from a university, so they can invite you to take part in a study to improve the way future incidents are dealt with | Definitely acceptable | 42 (4.2) | 70 (7) | | 63 (6.3) | 49 (4.9) | | 112 (5.6) |
| | Probably acceptable | 174 (17.4) | 206 (20.6) | 16.001** | 203 (20.3) | 177 (17.7) | 9.797* | 380 (19) |
| | Not sure | 221 (22.1) | 238 (23.8) | | 214 (21.4) | 245 (24.5) | | 459 (22.9) |
| | Probably not acceptable | 268 (26.8) | 229 (22.9) | | 230 (23) | 267 (26.7) | | 497 (24.9) |
| | Definitely not acceptable | 295 (29.5) | 257 (25.7) | | 290 (29) | 262 (26.2) | | 552 (27.6) |
| A journalist, so they can write a news article about the incident | Definitely acceptable | 17 (1.7) | 17 (1.7) | | 16 (1.6) | 18 (1.8) | | 34 (1.7) |
| | Probably acceptable | 42 (4.2) | 49 (4.9) | 2.439 | 43 (4.3) | 48 (4.8) | 4.709 | 91 (4.6) |
| | Not sure | 85 (8.5) | 100 (10) | | 84 (8.4) | 101 (10.1) | | 185 (9.3) |
| | Probably not acceptable | 143 (14.3) | 149 (14.9) | | 136 (13.6) | 156 (15.6) | | 292 (14.6) |
| | Definitely not acceptable | 713 (71.3) | 685 (68.5) | | 721 (72.1) | 677 (67.7) | | 1398 (69.9) |
| A charity such as the British Red Cross, so they can offer you support | Definitely acceptable | 72 (7.2) | 75 (7.5) | | 75 (7.5) | 72 (7.2) | | 147 (7.4) |
| | Probably acceptable | 277 (27.7) | 291 (29.1) | 1.183 | 270 (27) | 298 (29.8) | 3.131 | 568 (28.4) |
| | Not sure | 260 (26) | 245 (24.5) | | 254 (25.4) | 251 (25.1) | | 505 (25.3) |
| | Probably not acceptable | 205 (20.5) | 196 (19.6) | | 199 (19.9) | 202 (20.2) | | 401 (20.1) |
| | Definitely not acceptable | 186 (18.6) | 193 (19.3) | | 202 (20.2) | 177 (17.7) | | 379 (18.9) |
| A health-related government organisation, so they can send you a questionnaire to find out if you might benefit from extra care or support | Definitely acceptable | 108 (10.8) | 152 (15.2) | | 132 (13.2) | 128 (12.8) | | 260 (13) |
| | Probably acceptable | 370 (37) | 383 (38.3) | 11.036* | 361 (36.1) | 392 (39.2) | 2.502 | 753 (37.6) |
| | Not sure | 229 (22.9) | 208 (20.8) | | 225 (22.5) | 212 (21.2) | | 437 (21.9) |
| | Probably not acceptable | 155 (15.5) | 136 (13.6) | | 153 (15.3) | 138 (13.8) | | 291 (14.6) |
| | Definitely not acceptable | 138 (13.8) | 121 (12.1) | | 129 (12.9) | 130 (13) | | 259 (13) |
| A specialist NHS team, to provide you with information about ways to get support for any physical or mental health issues | Definitely acceptable | 258 (25.8) | 294 (29.4) | | 271 (27.1) | 281 (28.1) | | 552 (27.6) |
| | Probably acceptable | 438 (43.8) | 448 (44.8) | 6.897 | 440 (44) | 446 (44.6) | 0.978 | 886 (44.3) |
| | Not sure | 138 (13.8) | 126 (12.6) | | 139 (13.9) | 125 (12.5) | | 264 (13.2) |
| | Probably not acceptable | 87 (8.7) | 70 (7) | | 79 (7.9) | 78 (7.8) | | 157 (7.9) |
| | Definitely not acceptable | 79 (7.9) | 62 (6.2) | | 71 (7.1) | 70 (7) | | 141 (7.1) |

Continued

**Table 3** Continued

| Agency requesting personal information and reason for the request | | Frequency (%) believing if it is acceptable for the police to share for this reason | | | | | | |
|---|---|---|---|---|---|---|---|---|
| | | Shooting | Radiation | $\chi^2$ | Preincidents | Postincidents | $\chi^2$ | Total |
| A law firm, so they can offer to represent you in a 'no-win, no-fee' claim for compensation | Definitely acceptable | 31 (3.1) | 28 (2.8) | | 32 (3.2) | 27 (2.7) | | 59 (3) |
| | Probably acceptable | 76 (7.6) | 85 (8.5) | 2.796 | 83 (8.3) | 78 (7.8) | 6.699 | 161 (8.1) |
| | Not sure | 137 (13.7) | 128 (12.8) | | 127 (12.7) | 138 (13.8) | | 265 (13.3) |
| | Probably not acceptable | 166 (16.6) | 189 (18.9) | | 158 (15.8) | 197 (19.7) | | 355 (17.8) |
| | Definitely not acceptable | 590 (59) | 570 (57) | | 600 (60) | 560 (56) | | 1160 (58) |

*P<0.05; **P<0.01.
GP, general practitioner; NHS, National Health Service.

important also not to overstate the differences that we observed, however. Although statistically significant, in many cases these were small. Nonetheless, additional care might be required when communicating with the public about data sharing that is primarily intended to support a mental health response. It is possible that ongoing public campaigns to improve the understanding of mental ill-health may impact on these differences over time.

### Limitations
Several limitations with our data need to be considered. First, we restricted our sample to people aged 18–65 years. We are unable to extrapolate from our data to those older or younger than this. Second, although our sample was demographically representative of people aged 18–65 years in England, it may not have been psychologically representative of this population. In particular, it seems plausible that people who volunteer to answer internet surveys may be more accepting of data sharing than those who do not volunteer. If anything, our estimate of the number of people expressing reticence about data sharing is therefore likely to be an underestimate. Third, the scenarios used in our surveys were hypothetical. We hoped that repeating our survey soon after a series of terrorist attacks and a major disaster would make the issues involved more salient for participants. Doing this did not alter our results, possibly reflecting the fact that participants in our first survey were already thinking carefully about the scenarios that we presented. Nonetheless, it is possible that being directly involved in a major incident would alter someone's views about the desirability of data sharing. Fourth, for ethical reasons, we asked participants to consider that neither they nor anyone they knew well had been harmed in the scenarios that we described. This may have reduced the perceived need to participate

**Table 4** Attitudes towards proposed confidential database

| If I was caught up in a terrorist incident | | Frequency (%) of agreement | | | |
|---|---|---|---|---|---|
| | | Preincidents | Postincidents | $\chi^2$ | Total |
| I would want the police to give my contact details to the government organisation so that they can contact me about this service | Strongly agree | 209 (20.9) | 211 (21.1) | | 420 (21) |
| | Tend to agree | 454 (45.4) | 458 (45.8) | 3.918 | 912 (45.6) |
| | Neither | 177 (17.7) | 175 (17.5) | | 352 (17.6) |
| | Tend to disagree | 116 (11.6) | 97 (9.7) | | 213 (10.7) |
| | Strongly disagree | 44 (4.4) | 59 (5.9) | | 103 (5.2) |
| I would want my name to be included on this database | Strongly agree | 143 (14.3) | 135 (13.5) | | 278 (13.9) |
| | Tend to agree | 388 (38.8) | 366 (36.6) | 3.333 | 754 (37.7) |
| | Neither | 242 (24.2) | 265 (26.5) | | 507 (25.4) |
| | Tend to disagree | 156 (15.6) | 149 (14.9) | | 305 (15.3) |
| | Strongly disagree | 71 (7.1) | 85 (8.5) | | 156 (7.8) |
| I would be unhappy if my name was included on this database | Strongly agree | 89 (8.9) | 102 (10.2) | | 191 (9.6) |
| | Tend to agree | 181 (18.1) | 185 (18.5) | 2.470 | 366 (18.3) |
| | Neither | 305 (30.5) | 280 (28) | | 585 (29.3) |
| | Tend to disagree | 323 (32.3) | 321 (32.1) | | 644 (32.2) |
| | Strongly disagree | 102 (10.2) | 112 (11.2) | | 214 (10.7) |

These data were not split by scenario type, as it regarded a different scenario to that originally posed to participants.

**Table 5** Attitudes towards contact details being shared with another agency by the police

| Imagine the scenario we described occurred and the police decided to share your name and contact information with a health-related government organisation | | Frequency (%) of agreement | | | | | | |
|---|---|---|---|---|---|---|---|---|
| | | Shooting | Radiation | $\chi^2$ | Preincidents | Postincidents | $\chi^2$ | Total |
| I would be concerned that my details would be made public by accident | Strongly agree | 290 (29) | 239 (23.9) | | 259 (25.9) | 270 (27) | | 529 (26.4) |
| | Tend to agree | 360 (26) | 363 (36.3) | 10.004* | 365 (36.5) | 358 (35.8) | 0.742 | 723 (36.1) |
| | Neither | 201 (20.1) | 223 (22.3) | | 217 (21.7) | 207 (20.7) | | 424 (21.2) |
| | Tend to disagree | 130 (13) | 143 (14.3) | | 135 (13.5) | 138 (13.8) | | 273 (13.7) |
| | Strongly disagree | 19 (1.9) | 32 (3.2) | | 24 (2.4) | 27 (2.7) | | 51 (2.6) |
| I would trust the health organisation to keep my details secure | Strongly agree | 152 (15.2) | 161 (16.1) | | 171 (17.1) | 142 (14.2) | | 313 (15.7) |
| | Tend to agree | 363 (36.3) | 411 (41.1) | 8.537 | 375 (37.5) | 399 (39.9) | 7.741 | 774 (38.7) |
| | Neither | 266 (26.6) | 245 (24.5) | | 256 (25.6) | 255 (25.5) | | 511 (25.6) |
| | Tend to disagree | 164 (16.4) | 128 (12.8) | | 153 (15.3) | 139 (13.9) | | 292 (14.6) |
| | Strongly disagree | 55 (5.5) | 55 (5.5) | | 45 (4.5) | 65 (6.5) | | 110 (5.5) |
| It would be an invasion of my privacy | Strongly agree | 191 (19.1) | 159 (15.9) | | 164 (16.4) | 186 (18.6) | | 350 (17.5) |
| | Tend to agree | 255 (25.5) | 227 (22.7) | 15.770** | 246 (24.6) | 236 (23.6) | 2.549 | 482 (24.1) |
| | Neither | 306 (30.6) | 288 (28.8) | | 293 (29.3) | 301 (30.1) | | 594 (29.7) |
| | Tend to disagree | 204 (20.4) | 271 (27.1) | | 244 (24.4) | 231 (23.1) | | 475 (23.8) |
| | Strongly disagree | 44 (4.4) | 55 (5.5) | | 53 (5.3) | 46 (4.6) | | 99 (5) |
| I would be concerned about how my information might be used in the future | Strongly agree | 297 (29.7) | 251 (25.1) | | 266 (26.6) | 282 (28.2) | | 548 (27.4) |
| | Tend to agree | 394 (39.4) | 414 (41.4) | 6.237 | 405 (40.5) | 403 (40.3) | 1.755 | 808 (40.4) |
| | Neither | 177 (17.7) | 203 (20.3) | | 192 (19.2) | 188 (18.8) | | 380 (19) |
| | Tend to disagree | 107 (10.7) | 109 (10.9) | | 115 (11.5) | 101 (10.1) | | 216 (10.8) |
| | Strongly disagree | 25 (2.5) | 23 (2.3) | | 22 (2.2) | 26 (2.6) | | 48 (2.4) |
| I would be concerned that my employer would be told information about me | Strongly agree | 116 (11.6) | 109 (10.9) | | 115 (11.5) | 110 (11) | | 225 (11.3) |
| | Tend to agree | 232 (23.2) | 227 (22.7) | 0.847 | 228 (22.2) | 231 (23.1) | 0.738 | 459 (45.9) |
| | Neither | 281 (28.1) | 298 (29.8) | | 284 (28.4) | 295 (29.5) | | 579 (28.9) |
| | Tend to disagree | 261 (26.1) | 260 (26) | | 267 (26.7) | 254 (25.4) | | 521 (26.1) |
| | Strongly disagree | 110 (11) | 106 (10.6) | | 106 (10.6) | 110 (11) | | 216 (10.8) |
| I would be concerned that my details might be shared by the health organisation with other groups without my permission | Strongly agree | 273 (27.3) | 233 (23.3) | | 245 (24.5) | 261 (26.1) | | 506 (25.3) |
| | Tend to agree | 384 (38.4) | 380 (38) | 8.629 | 382 (38.2) | 382 (38.2) | 2.127 | 764 (38.2) |
| | Neither | 201 (20.1) | 245 (24.5) | | 227 (22.7) | 219 (21.9) | | 446 (22.3) |
| | Tend to disagree | 111 (11.1) | 118 (11.8) | | 114 (11.4) | 115 (11.5) | | 229 (11.5) |
| | Strongly disagree | 31 (3.1) | 24 (2.4) | | 32 (3.2) | 23 (2.3) | | 55 (2.8) |
| I would want to be kept informed about how my information was being used | Strongly agree | 509 (50.9) | 489 (48.9) | | 518 (51.8) | 480 (48) | | 998 (49.9) |
| | Tend to agree | 340 (34) | 357 (35.7) | 9.692* | 339 (33.9) | 358 (35.8) | 5.162 | 697 (34.9) |
| | Neither | 114 (11.4) | 128 (12.8) | | 109 (10.9) | 133 (13.3) | | 242 (12.1) |
| | Tend to disagree | 24 (2.4) | 24 (2.4) | | 27 (2.7) | 21 (2.1) | | 48 (2.4) |
| | Strongly disagree | 13 (1.3) | 2 (0.2) | | 7 (0.7) | 8 (0.8) | | 15 (0.8) |
| It would not bother me at all | Strongly agree | 58 (5.8) | 72 (7.2) | | 64 (6.4) | 66 (6.6) | | 130 (6.5) |
| | Tend to agree | 204 (20.4) | 222 (22.2) | 7.131 | 202 (20.3) | 224 (22.4) | 2.232 | 426 (21.3) |
| | Neither | 308 (30.8) | 330 (33) | | 332 (33.2) | 306 (30.6) | | 638 (31.9) |
| | Tend to disagree | 270 (27) | 245 (24.5) | | 257 (25.7) | 258 (25.8) | | 515 (25.8) |
| | Strongly disagree | 160 (16) | 131 (13.1) | | 145 (14.5) | 146 (14.6) | | 291 (14.6) |
| I would be reassured that a health organisation was looking out for me | Strongly agree | 107 (10.7) | 148 (14.8) | | 137 (13.7) | 118 (11.8) | | 255 (12.8) |
| | Tend to agree | 425 (42.5) | 427 (42.7) | 12.829* | 415 (41.5) | 437 (43.7) | 7.791 | 852 (42.6) |
| | Neither | 305 (30.5) | 302 (30.2) | | 319 (31.9) | 288 (28.8) | | 607 (60.7) |
| | Tend to disagree | 112 (11.2) | 79 (7.9) | | 91 (9.1) | 100 (10) | | 191 (9.6) |
| | Strongly disagree | 51 (5.1) | 44 (4.4) | | 38 (3.8) | 57 (5.7) | | 95 (4.8) |

Continued

**Table 5**   Continued

| Imagine the scenario we described occurred and the police decided to share your name and contact information with a health-related government organisation | | Frequency (%) of agreement | | | | | | |
|---|---|---|---|---|---|---|---|---|
| | | Shooting | Radiation | $\chi^2$ | Preincidents | Postincidents | $\chi^2$ | Total |
| I would be concerned that my GP might be told | Strongly agree | 44 (4.4) | 42 (4.2) | | 41 (4.1) | 45 (4.5) | | 86 (4.3) |
| | Tend to agree | 102 (10.2) | 89 (8.9) | 2.679 | 90 (9) | 101 (10.1) | 3.554 | 191 (9.6) |
| | Neither | 296 (29.6) | 277 (27.7) | | 292 (29.2) | 281 (28.1) | | 573 (28.7) |
| | Tend to disagree | 369 (36.9) | 397 (39.7) | | 397 (39.7) | 369 (36.9) | | 766 (38.3) |
| | Strongly disagree | 189 (18.9) | 195 (19.5) | | 180 (18) | 204 (20.4) | | 384 (19.2) |

*P<0.05; **P<0.01.
GP, general practitioner.

in a psychological screening programme and hence the perceived acceptability of data sharing. People who are injured, bereaved or otherwise harmed in an incident may be more willing for their data to be shared. Finally, we have no information on our participants' previous experiences of data sharing or data protection breaches. Prior experience of breaches, from whatever source or agency, may make people more reluctant to consider future data sharing. This may be relevant for the minority of people in our sample who were reluctant to share data. It would also reinforce the need for clear information and reassurance about data handling.

## CONCLUSION

Disagreement often exists between officials and agencies about the appropriateness of, and best mechanisms for, data sharing in the aftermath of a major incident. These disagreements are also apparent among members of the public: while most support the sharing of data between the police and health services in order to facilitate public health measures, this is by no means a unanimous view. The absence of a clear consensus reinforces the view that agencies should consider the benefits and risks of sharing data with organisations that seek to protect their mental health. In order to assuage public concerns, agencies who hold such data should communicate to the public their policies for securely holding the data and for updating people on how the data will be used.

**Contributors**  GJR developed the idea for the survey and the survey materials in conjunction with ANR, RA, NiG and NeG, Ipsos MORI carried out the data collection. RW carried out the data analysis. GJR drafted the manuscript and revised the manuscript with comments from all authors. GJR accepts full responsibility for the work, had access to the data and controlled the decision to publish.

**Funding**  The research was funded by the National Institute for Health Research Health Protection Research Unit (NIHR HPRU) in Emergency Preparedness and Response at King's College London in partnership with Public Health England (PHE), and by the NIHR HPRU in Evaluation of Interventions at the University of Bristol in partnership with PHE. HPRU-2012-10414.

**Disclaimer**  The views expressed are those of the author(s) and not necessarily those of the NHS, the NIHR, the Department of Health or Public Health England.

**Competing interests**  None declared.

**Patient consent**  Not required.

**Ethics approval**  The study was approved by the Psychiatry, Nursing and Midwifery Research Ethics Committee at King's College London (ref: HR-16/17-3814). All participants provided informed consent before starting the surveys.

**Provenance and peer review**  Not commissioned; externally peer reviewed.

**Data sharing statement**  No additional data available.

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
