## [Reviewer comments · BMJ Open]

ARTICLE DETAILS

TITLE (PROVISIONAL)	Public attitudes in England towards the sharing of personal data following a mass casualty incident: A cross-sectional study
AUTHORS	Rubin, GJ; Webster, Rebecca; Rubin, Antonia; Amlot, Richard; Grey, N; greenberg, neil;

VERSION 1 – REVIEW

REVIEWER	Itamar Ashkenazi Hillel Yaffe Medical Center, Israel
REVIEW RETURNED	22-Mar-2018

GENERAL COMMENTS	Title: Public attitudes towards the sharing of personal data following a mass casualty incident: A cross-sectional study (bmjopen-2018-022852) Methods: Internet interview Authors' findings: This study is about how people perceive sharing of information concerning their identity by the police with health and other institutions following a situation imposing possible health risks including mental well-being. Two scenarios are presented representing mass exposure to mass-shooter attack and a potentially hazardous substance. Reviewer's comments: This is a well-written study with an interesting methodology in a virtual scenario that has become more and more relevant in our daily reality. For this reason, I suggest the editors to consider accepting this manuscript for publication. I have only three comments for the authors: In this study, responders answering the mass-shooting scenario questions were different from those answering the radiation scenario questions. More people were willing to have information shared by the police following a radiation event compared to a mass-shooting event. Though differences were statistically meaningful, results are only 4-5% apart, questioning whether these differences are important. With 2000 participants who are not matched, even small differences between the groups will achieve the $p < 0.05$ threshold for statistical significance.
--

	The authors do not discuss why no difference was found between the "before" and "after" questionnaires. The authors do not discuss the fact that only 6% were willing to answer the questions "after" compared to 10% "before". This could affect the validity of the findings found in the second round of questions. How do the researchers explain the discrepancy between the observation that 15.8% would not want their contact details to be shared by the police in order to allow screening while 27.9% (almost twice as much) reported they would be unhappy if their names were included in such a list? Does this show some kind of limitation of the method used for this survey?
--	--

REVIEWER	Chris Brewin Professor of Clinical Psychology, University College London, U.K.
REVIEW RETURNED	11-Apr-2018

GENERAL COMMENTS	This is an important and original article with a clear public policy rationale. The main change I would like to see is to present data on the original 5-point scales in Tables 2-5. From a public policy perspective I believe it will be more informative to identify individuals who "strongly disagreed" versus simply "disagreed" or who considered options to be "definitely unacceptable" versus "probably unacceptable", etc. Given that people selecting the milder preferences are closer to the midpoint and have opted not to endorse the stronger preferences, I would make these more extreme responders the focus in the Abstract and Discussion. Whether or not the chi-squared analyses should be changed is a moot point. I would have no objection to them being based on the combined categories as they are now. The other comment concerns the limitations. In the scenario respondents are told that neither they nor anyone they know have been harmed. This may have been taken to mean that they were not harmed either mentally or physically. Whereas people are aware of the insidious effects of radiation that may not be immediately apparent, fewer may be aware of mental health consequences that only become apparent later. This could suppress enthusiasm for the various health initiatives suggested. I suggest the authors discuss this point in slightly greater detail, acknowledging that drawing attention to the potential mental consequences might have affected the results.
---

VERSION 1 – AUTHOR RESPONSE

Reviewer: 1

1. In this study, responders answering the mass-shooting scenario questions were different from those answering the radiation scenario questions. More people were willing to have information shared by the police following a radiation event compared to a mass-shooting event. Though differences were statistically meaningful, results are only 4-5% apart, questioning whether these differences are important. With 2000 participants who are not matched, even small differences between the groups will achieve the $p < 0.05$ threshold for statistical significance.

We have added a caveat to the relevant section of the discussion noting that “It is important also not to overstate the differences that we observed – although statistically significant, in many cases these differences were small.”

2. The authors do not discuss why no difference was found between the "before" and "after" questionnaires. The authors do not discuss the fact that only 6% were willing to answer the questions "after" compared to 10% "before". This could affect the validity of the findings found in the second round of questions.

In practice, the difference in response rate is probably due to the timing of the second survey, which occurred during the English Summer holidays. It is unlikely to reflect a reduced interest in the topic of the survey, which might in turn contribute to response bias. We have added a line about the lower response rate to the first paragraph of the results section.

We feel that the most parsimonious explanation for the lack of any difference between the pre- and post-incident surveys is that participants in the pre-incident survey were already thinking carefully about our scenarios, hence the media focus on the acts of terrorism and tower block fire did not alter responses in the second survey. We have now included a line in the limitations section which offers this as an explanation.

3. How do the researchers explain the discrepancy between the observation that 15.8% would not want their contact details to be shared by the police in order to allow screening while 27.9% (almost twice as much) reported they would be unhappy if their names were included in such a list? Does this show some kind of limitation of the method used for this survey?

The questions ask about two separate components of the service. While the first asks whether the police can share details with the government organisation “so that they can contact me about this service” the second asks about having one’s details included on a database. It seems reasonable that people are more accepting of being offered information about a service (and then possibly being able to opt in / out) vs being included on a database.

Reviewer: 2

Reviewer Name: Chris Brewin

Institution and Country: Professor of Clinical Psychology, University College London, U.K.

Competing Interests: None declared

1. This is an important and original article with a clear public policy rationale. The main change I would like to see is to present data on the original 5-point scales in Tables 2-5. From a public policy perspective I believe it will be more informative to identify individuals who "strongly disagreed" versus simply "disagreed" or who considered options to be "definitely unacceptable" versus "probably unacceptable", etc. Given that people selecting the milder preferences are closer to the midpoint and have opted not to endorse the stronger preferences, I would make these more extreme responders the focus in the Abstract and Discussion. Whether or not the chi-squared analyses should be changed is a moot point. I would have no objection to them being based on the combined categories as they are now.

We have now changed tables 2 to 5 to include all response options. We have recalculated the chi-squared tests, which have changed some results but only marginally – the overall pattern has not changed. We have also adjusted some of the text in the results section as a consequence.

We have changed the first paragraph of the discussion and the abstract to include data for the more extreme responders.

2. The other comment concerns the limitations. In the scenario respondents are told that neither they nor anyone they know have been harmed. This may have been taken to mean that they were not harmed either mentally or physically. Whereas people are aware of the insidious effects of radiation

that may not be immediately apparent, fewer may be aware of mental health consequences that only become apparent later. This could suppress enthusiasm for the various health initiatives suggested. I suggest the authors discuss this point in slightly greater detail, acknowledging that drawing attention to the potential mental consequences might have affected the results.

While we agree that this might have suppressed enthusiasm for the mental health screening programme proposed in the survey, we are not sure if the issue represents a limitation per se. Instead, it seems to reflect a valid reason underlying the beliefs and attitudes that were reported by our participants. We have therefore added an additional line to the second paragraph of our discussion which notes that “while people may be familiar with the concept of physical health effects that develop some time later after a radiation incident, they may be less familiar with the concept of mental health effects that only become apparent later: this lack of understanding may also have reduced the perceived desirability of a mental health screening programme across both conditions.”

VERSION 2 – REVIEW

REVIEWER	Chris Brewin University College London, UK
REVIEW RETURNED	26-Apr-2018
GENERAL COMMENTS	The authors have responded satisfactorily to my previous comments
REVIEWER	Itamar Ashkenazi Hillel Yaffe Medical Center
REVIEW RETURNED	28-Apr-2018
GENERAL COMMENTS	Thank you for allowing me to review the revised manuscript “Public attitudes towards the sharing of personal data following a mass casualty incident: A cross-sectional study (bmjopen-2018-022852)”. This is a well-written study with an interesting methodology in a virtual scenario that has become more and more relevant in our daily reality. For this reason, I suggested the editors to consider accepting this manuscript for publication. I had only three comments for the authors and they clarified my misunderstandings as they revised the manuscript accordingly. I have no further comments.